# Predicting Perceived Problems in Self-Administered 24-Hour Dietary Recalls: A Quantitative Think-Aloud Study Comparing Automated Self-Assisted 24-Hour Dietary Assessment Tool (ASA24^®^) and INTAKE24© in University Students

**DOI:** 10.3390/nu14204281

**Published:** 2022-10-13

**Authors:** Katlyn M. Mackenzie, Deborah A. Kerr, Clare Whitton, Zenobia Talati, Tracy A. McCaffrey, Barbara A. Mullan

**Affiliations:** 1Curtin School of Population Health, Curtin University, Kent Street, Perth 6845, Australia; 2Enable Institute, Curtin University, Kent Street, Perth 6845, Australia; 3Curtin Health Innovation Research Institute, Curtin University, Kent Street, Perth 6845, Australia; 4Department of Nutrition and Dietetics, Monash University, 264 Ferntree Gully Road, Notting Hill 3168, Australia

**Keywords:** 24HR dietary recall, self-report error, INTAKE24, ASA24, food habits, mindful eating, dietary assessment technology

## Abstract

Demographic and psychosocial factors concerning dietary assessment error have been explored, but few studies have investigated the perceived problems experienced when completing dietary recalls. The aim of this research was to (i) compare the perceived problems encountered in two commonly used self-administered 24-hour dietary recall (24HR) programs (INTAKE24© and ASA24^®^) and (ii) explore whether mindful and habitual eating are associated with perceived problems during dietary recall. A randomised quantitative crossover design and think-aloud methodology were employed. Undergraduate university students (N = 55, *M_age_* = 25.5, *SD* = 8.2, 75% female) completed a food habits and mindfulness questions pre-program, one 24HR (whilst thinking aloud), and a systems usability scale post-program. A week later, they completed the other 24HR (whilst thinking aloud). During a pilot, a coding frame of perceived problems was devised to quantify participants’ perceived problems. INTAKE24© generated significantly fewer perceived problems across all categories compared to ASA24^®^ (17.2 vs. 33.1, *p <* 0.001). Of the participants, 68% reported a preference for INTAKE24© over ASA24^®^. Hierarchical multiple regression showed that habits and systems usability were significant predictors of perceived problems for INTAKE24© only. No significant predictors were found for ASA24^®^. The results provide insight into perceived problems people may encounter when using 24HR tools.

## 1. Introduction

Chronic disease related to the consumption of a poor diet is currently the leading source of disability and death globally [1]. Making dietary recommendations to promote healthy eating requires accurate data on population dietary intake [2]. However, various factors influence accuracy, and, as with any self-reported data, dietary intake information is vulnerable to measurement error [3]. It is essential to understand how and why these errors occur to improve the reliability of population dietary surveillance and epidemiology [4].

There are multiple self-report methods used to capture dietary intake, with the 7-day weighed dietary record designed to capture more detailed information about food and beverages consumed [3]. Whilst cognitive difficulty is low for dietary records, participant burden is high and can contribute to misreporting of total energy intake [5]. As such, there has been a move towards the use of multiple non-consecutive 24-hour dietary recalls for large-scale population surveillance [6]. Most national surveys, such as the National Health and Nutrition Examination Survey (NHANES) in the United States of America (USA), include interviewer-administered Automated Multiple-Pass Method 24-Hour Recalls (24HR) within their population surveillance [7]. The emergence of self-administered web-based 24HR aims to reduce cost and address participant burden and acceptability. Participants are guided through their dietary recall using a series of prompts (e.g., time of consumption) and standard images of foods to help estimate portion sizes [8]. Web-based 24HR minimises response error by considering spelling mistakes and having a simple-to-use database and multiple prompts (i.e., reminders of commonly forgotten foods) [8]. Two widely used self-administered web programs, INTAKE24© and the Automated Self-Administered 24HR dietary assessment tool (ASA24^®^) [9,10] capture participants’ consumed food and beverages by guiding them through a structured recall from the previous 24 h, with images for portion estimation [11]. Understanding contributions to reporting errors is essential to the improvement of these self-report methods [12]. 

Previous research on reporting errors in dietary recalls has primarily focused on participants’ characteristics, such as age, gender, socioeconomic status, and body mass index (BMI) [4,13]. For example, Livingstone and Black’s [5] systematic review found that individuals who were high in BMI and health-conscious, of older age and female, and low socio-educational status, were more likely to underreport their energy intake. Other literature has argued that psychological characteristics, such as emotional influence [14], fear of negative evaluation, food restraint [15], and body perceptions [16], pose significant issues in self-reporting accuracy. 

Compared to a more routine diet, dietary recall errors may occur when eating habits are varied or irregular [17]. For example, Osadchiy et al. [11] tested whether progressive recall (recalling multiple times a day) was more accurate than a standard 24HR, completing follow-up interviews with participants on their experience. The study found that having a greater meal variety made it more difficult for participants to remember their intake when undertaking the 24HR than when they had more regular eating routines. Additionally, the study found no significant differences in the accuracy between the two recall methods, with 65% of participants believing the 24HR better fit into their daily routines [11]. 

Though studies have not previously found an association between habitual diets and recall accuracy, others have investigated repetition and recall in other memory research areas and found contrasting positive and negative [18,19] associations. The negative repetition effect suggests that multiple presentations of the same stimulus can negatively affect memory recall [19]. As such, someone who eats the same daily snacks may not remember them as they are consumed automatically. Conversely, as habits are repetitive, they can enhance routine and limit memory error, as remembering completed tasks may become second nature [20]. 

Furthermore, paying attention can reduce errors in memory recall [21]. For example, Higgs Rodero [22] investigated the effect of manipulating attention during food consumption and concluded that memory of meals was impaired by distractions, indicating the importance of attention for accurate dietary recall. Attentive eating (mindful eating) helps participants to engage in and be aware while eating and has previously been measured by mindfulness eating questionnaires [23]. For example, Higgs and Donohoe [24] found that those who ate mindfully had a more vivid memory of the meal later that day. Therefore, habit and mindfulness may contribute to how participants perceive problems during 24HR, but this has not yet been explored.

With most research designs, researchers cannot understand participants’ thoughts during self-report or the specific problems they may encounter. Think-aloud is an observational method wherein participants verbalise all thoughts whilst completing a given task and engage in real-time feedback of thoughts and answers [25]. In particular, the measure was designed to elicit present-time feedback on perceived problems, misreporting, and responses of emotion when working through questionnaires [25]. For example, Razali et al. [26] used the methodology to determine which teaching experience students most benefitted from. No previous research has explored how habitual behaviour and mindfulness are involved in perceived problems during 24-hour dietary recall. Therefore, the aim of this research was to (i) compare the perceived problems encountered in two commonly used dietary recall programs and (ii) to explore whether higher levels of mindful and habitual eating contributed to fewer perceived problems during dietary recall, as made salient via differences in think-aloud data.

## 2. Materials and Methods

### 2.1. Design

A randomised quantitative crossover design was used in the current study and was conducted from August 2021 to September 2021. The study employed a think-aloud methodology [27]. The central measures of the study were the number of perceived problems raised when using INTAKE24© and ASA24^®^ [9,10].

### 2.2. Participants

Recruitment for the main study consisted of a convenience sample of undergraduate psychology students (years 1 to 3) at Curtin University, recruited via the SONA participant pool. SONA is an online student research participation system created for undergraduate psychology students, whereby students are required to achieve five face-to-face SONA ‘points’ each semester. SONA points cannot be exchanged for money, nor do they have any monetary value. There were no other exclusion criteria. 

### 2.3. Procedure 

After ethics approval from the Curtin University Human Research Ethics Committee (HREC-2021-0295), recruitment commenced. The study was advertised to the participant pool, inviting them to participate. Information provided to participants explained that their appointment times were to be the same for both weeks of participation. When the participants signed up, they were provided with the study information and consent form. To link the separate timeslot data and de-identify participants, they were asked to create an unidentifiable unique ID and provide their student ID. Questionnaires were administered via Qualtrics.com (copyright version August–September 2021, Provo, UT) [28] and personalised logins and links for INTAKE24© and ASA24^®^ were created. Once consent to participate was provided, participants were asked to complete demographic information (i.e., age, identified gender, self-reported height, and weight), Creatures of Habit Scale (See Appendix A), and Mindfulness Eating Questionnaire (see Appendix A), taking approximately 10 min. Height and weight were calculated for BMI categories (BMI cut-offs: Underweight ≤18.49 kg/m^2^, Healthy Weight 18.5 to 24.9 kg/m^2^, Overweight 25 to 29.9 kg/m^2^, and Obese ≥30 kg/m^2^) [29]. Participants reported to the Psychology Experimental Research Laboratories at Curtin (PERL-C) at their chosen timeslot, where they were provided with a brief study description explaining the think-aloud design. The 24HR programs were randomised to reduce order effects, whereby half completed ASA24^®^ as their first recall, and half completed INTAKE24© as their first recall. After any questions had been answered, participants were asked to complete the first 24HR whilst thinking aloud. The researcher exited the room once the audio recording began. After the 24HR was complete, participants were asked to complete the Systems Usability Scale (see Appendix A) via Qualtrics.com (copyright version August–September 2021, Provo, UT) [28] for the 24HR program. Participants were asked to knock on the door to let the researcher know they were finished, and the recording was stopped. A limited debrief was provided to remind participants of their second timeslot and allow any questions about their 24HR to be answered. No information was given about the requirements for the second timeslot so as not to influence participants’ responses. A week later, participants returned for their second timeslot and followed the same procedure but completed the second 24HR program and usability questions. A single *forced* question was asked in the final recall as to which tool they preferred (INTAKE24© or ASA24^®^), followed by a final debrief. Each timeslot to complete the 24HR took approximately 30 min. Students were awarded five SONA points for participation after completion of both timeslots. 

### 2.4. Measures 

#### 2.4.1. 24-Hour Recall Programs (INTAKE24© and ASA24^®^)

Participants used the computer-based programs INTAKE24© and ASA24^®^ to recall what they had consumed in the previous 24 h. Both methods are based on the USDA Automated Multiple-Pass Method (AMPM), which uses a five-step approach to enhance completion and accuracy when collecting dietary data [30]. The AMPM method begins by asking participants to list all consumed items in an unstructured way. Over the next three steps, the program structurally selects each item recalled in the first step and asks specific questions to probe participants’ memory on the portion size, amount consumed, brand, and cooking method of each item. The final step asks participants if any items may have been missed or forgotten, using time frames without items as a memory probe [30]. ASA24^®^ is an easy-to-use, low-cost, self-administered 24HRtool and was one of the first AMPMs, developed by the National Cancer Institute, Bethesda, MD, with validation amongst adults and children (ASA24^®^-Kids) [3,12]. Similarly, INTAKE24© is an open-source, self-completed, computerised 24HR that includes placeholders for various meals and snacks. The system was designed by a multidisciplinary team using an iterative process informed by user testing with 11–24-year-old users at Newcastle University, with the purpose of data collection in national food and nutrition surveys [9,12]. The current study used the Australian-adapted INTAKE24© (https://intake24.com/; accessed from July – August 2021) and ASA24^®^ to permit usage of the food composition data from AUSNUT 2011-12 (accessed from July – August 2021) [31]. 

#### 2.4.2. Creatures of Habit Scale

The Creatures of Habit Scale is a 27-item questionnaire designed to measure habitual and routine behaviour and is made up of two factors (i.e., routine and automaticity) combined into an overall habit score [32]. The routine component has 16 questions, and automaticity has 11 questions. An example item includes, “I generally eat the same things for breakfast every day”. Only the questions related to food (n = 15) were used (see Appendix A). Given this, people with higher scores on these food-related questions had stronger food habits. Participants were required to rate their food habits on a five-point Likert scale from 1 (strongly disagree) to 5 (strongly agree), with total scores ranging from 15 to 75. Ersche et al. [32] found the full scale to be a valid and reliable measure for assessing habitual behaviour (Cronbach alpha (*α*)) for routine (*α* = 0.89) and automaticity (*α* = 0.86). The food-only questions yielded excellent reliability for the current study’s sample (*α* = 0.86).

#### 2.4.3. Mindfulness Eating Questionnaire

The 20-item, two-factor abbreviated Mindfulness Eating Questionnaire was used to measure mindful eating (see Appendix A). This questionnaire is made up of two factors (i.e., awareness and recognition) and the results are combined into an overall mindful eating score [23]. The awareness component has 11 questions and recognition has 9 questions (i.e., if they felt hungry or full). Participants were asked to choose the option that best describes how they feel when they eat. An example item includes “I taste every bite of food I eat”. The Mindfulness Eating Questionnaire uses a four-point Likert Scale from 1 (not representative) to 4 (representative), with scores ranging from 20 to 80, with scores nearing 80 representing greater mindful eating. Previous validation of the scale found it to be a reliable measure of mindful eating for awareness (*α* = 0.75) and recognition (*α* = 0.83) [23]. The current study’s sample yielded good reliability across awareness and recognition (*α* = 0.71).

#### 2.4.4. Think-Aloud

Ericsson and Simon first explored the think-aloud methodology in 1980 [27]; however, the instructions that were provided to participants in the present ‘think-aloud’ methodology were adapted from more recent work by Darker and French [33] and French et al. [34]. Briefly, participants were asked to verbalise all thoughts that came to mind when completing the 24HR. To reduce any influence, the researcher was not in the same room as the participant. As a reminder, a paper printout stating “Please Remember to Think-Aloud” was placed at the top of the computer screen. A convenience sample (*n* = 7) of five undergraduate students and two members of the public was recruited for the pilot. The pilot audio recordings were transcribed and analysed to observe any perceived problems in the procedure and develop a standard checklist of participants’ perceived problems for use in the main study (See Appendix A). Usability evaluators may encounter problems due to the methodology’s unnatural nature, which may cause participants to alter their answers or provide descriptive data rather than explanatory data. This has previously been addressed by allowing test participants to perform the task silently and comment afterwards on their performance (known as retrospective think-aloud; RTA) [27]. However, the study’s objective was to observe the problems raised using the two 24HR programs and retaining retrospective memory following a cognitively demanding food recall may provide inaccurate results [35,36]. Earlier work by Alhadreti [36] showed that concurrent think-aloud (CTA) outperformed RTA during usability testing, with participants reporting more problems during CTA, providing validation for its use in the study.

#### 2.4.5. Systems Usability Scale

The Systems Usability Scale is a fast and easy way to measure how usable participants find a specific system [37]. The scale’s specific measurements include system efficiency, effectiveness, and satisfaction graded on a five-point Likert scale, scoring from 0 to 100 (see Appendix A). Based on Bangor et al.’s. [37] ‘university grade analogy’, scores closest to 100 were considered ‘superior’, scores less than 70 were considered ‘in need of improvement’, and scores of 50 or less were a ‘usability failure’. Participants were asked questions from the original Systems Usability Scale statements to measure the usability of both INTAKE24© and ASA24^®^ [37]. Additionally, the word “cumbersome” in item 8 was replaced with a more commonly known synonym (i.e., “awkward”), as was demonstrated to be effective in Bangor et al. [37]. Past validation found excellent reliability (*α* = 0.91) [37]. In the current study, good reliability was found for ASA24^®^ (*α* = 0.88) and INTAKE24© (*α* = 0.86).

#### 2.4.6. Program Preference

Program preference was measured using a singular question, “Taking into account both 24-hour food recalls, please indicate which program you preferred”, with two choices: “Timeslot 1” or “Timeslot 2”. As timeslots 1 and 2 differed for each participant, the results were combined against each unique ID to ensure their preferred timeslot choice aligned with their randomised recall order. The data were then dummy coded with preference (1) indicating that the participant preferred INTAKE24© and (2) ASA24^®^.

### 2.5. Sample Size Calculation

Previous research on 24HR studies and the think-aloud methodology is limited, though a sensitivity analysis of a similar study yielded effects of a medium range [33]. Using G*Power 3.1 (Heinrich-Heine University, Düsseldorf, Germany) [38], a minimum of 57 participants were determined to be sufficient to detect a medium effect size (*f* = 0.19, and 95% confidence interval) with a power of 0.80 and an alpha level of 0.05. Given this, a total of 65 undergraduate students gave informed consent and voluntarily signed up to participate in this face-to-face study. Of those who signed up, six participants did not attend their timeslots, three did not complete all the required information, and one forgot to think aloud. Thus, they were excluded from the analyses, leaving 55 participants. After removing participants who did not complete all requirements, a missing-values analysis was run across all relevant variables. No participant answers were identified as missing; thus, expectation maximisation for missing values was unnecessary. 

### 2.6. Assumptions

Before interpreting the results, assumptions and normality were checked and, after visual inspection, deemed to be normal. Short Likert-scale lengths are sensitive to skewness [39]; thus, our study focused on the summation of items (i.e., total habit score, total mindfulness score, total systems usability score), so univariate normality was deemed to be interpretable. There were no multivariate outliers as Mahalanobis distance was below the critical *χ*^2^ of 16.27 (*α* = 0.001) for three predictor variables (for any cases) in the data, indicating no concern for multivariate outliers [40]. Stem-and-leaf plots and boxplots indicated each variable’s normal distribution in the regression without univariate outliers. After visually assessing the normal probability plot of standardised residuals and the scatterplot of standardised residuals against standardised predicted values, it was determined that normality, linearity, and homoscedasticity of residuals were acceptable. Finally, relatively high tolerance for all three predictors in the final regression model determined that multicollinearity would not interfere with the interpretation of the multiple regression analysis.

### 2.7. Data Analyses

Before testing the hypotheses, a bivariate correlation was run to analyse the associations between all target variables. The frequency of overall perceived problems within each recall program and the different perceived problems for each program were calculated. To test the primary objective of the study, a paired-samples *t*-test was performed to determine the differences in perceived problems between ASA24^®^ and INTAKE24© (Analysis 1). To further explain differences, multiple paired-samples *t*-tests were run to determine what types of perceived problems were most common in each 24HR program, including between and within perceived problem categories for both programs. The results were then compared to participants’ preferred programs to see if the difference between the perceived problems in each program aligned with the most preferred program. Secondarily, a hierarchical multiple regression analysis was conducted to test whether food habits, and mindful eating significantly predicted participants’ number of perceived problems in ASA24^®^ and INTAKE24© whilst controlling for systems usability (Analysis 2). Cohen’s conventions were used to determine the effect size for each analysis [41]. All analyses were run using SPSS (IBM SPSS Statistics for Windows, Version 28.0. Armonk, NY, USA) [42].

## 3. Results

### 3.1. Think-Aloud Checklist

Think-aloud data were coded quantitatively via an exploratory checklist using four of the seven participants during the pilot run. Four audio recordings were transcribed and coded by separate researchers (K.M. and D.K.) to create the checklist used to explore any perceived problems participants encountered during their recall. Both researchers discussed any disagreements and revised the coding framework. An additional six audio recordings from the main study were checked against the checklist to test inter-rater reliability. This validation method was based on a previous study by Darker and French [33]. A reasonable degree of reliability was found initially between the two raters (*k* = 0.60). After further discussion, a checklist was agreed upon and deemed to have perfect reliability (*k* = 1) [43]. Given the reliability, the quantifiable categories of ‘perceived problems’ raised during the think-aloud task were used on the remaining sample (*N* = 55) using the final checklist (See Appendix A).

The final coding frame comprised perceived problems with (1) remembering, (2) the program, (3) emotions (i.e., perceived emotional responses), and (4) no perceived problems. Category (1) consisted of perceived problems with remembering (a) consumption, (b) portions, (c) items, and (d) guessing from memory. Category (2) consisted of perceived problems with (a) what to input, (b) where to input, and (c) individual perceived problems with the program. Category (3) consisted of perceived emotions of (a) frustration and (b) confusion. Examples of quotes from participants are identified in Table 1.

### 3.2. Demographics

The final sample included 75% female and 25% male university student participants aged 18–49 years (*M* = 25.56, *SD =* 8.2). Means (*M*), standard deviations (*SD*), and descriptive statistics for food habits and mindfulness scores are displayed in Table 2. 

### 3.3. Preliminary Analysis

Intercorrelations from the bivariate correlation were analysed to determine the associations between the target variables (Table 3). Overall, perceived problems were significantly associated with all categories of problems (*p* < 0.05). Separate analyses for each problem category were not conducted, given the significantly large associations and overlap of problems (i.e., participants experienced memory problems that were also accompanied by emotion problems).

### 3.4. Analysis 1

On average, participants experienced 33.1 (*SD* 17.4) problems while completing ASA24^®^ and 17.2 (*SD* 11.3) problems while completing INTAKE24© (*p* < 0.001). On average, participants had 15.8, 95% CI [−19.8, −11.9] fewer perceived problems using INTAKE24© which was 1.9× fewer compared to ASA24^®^
*p* < 0.001. Differences between and within problem categories (i.e., discrimination within each program’s perceived problem categories) for both programs are displayed in Table 4. Overall, INTAKE24© was the most preferred program, with participants reporting fewer perceived remembering (8.8), program (3.0), and emotion problems (4.0; *p* < 0.001; Table 4). 

### 3.5. Analysis 2

#### 3.5.1. ASA24^®^

Hierarchical multiple regression showed that no single predictor was statistically significant (Table 5.) The systems usability score accounted for a nonsignificant 3.1% of variance, *R*^2^ = 0.03, *F*(1, 53) = 1.68, *p* = 0.200, in perceived problems. This was followed by food habits and mindfulness, which accounted for an additional nonsignificant 6.8% of variance, *R*^2^_change_ = 0.07, *F_change_*(2, 51) = 1.93, *p* = 0.156 with a ‘small to medium’ effect (*f*^2^ = 0.11) [41].

#### 3.5.2. INTAKE24©

A hierarchical multiple regression showed that higher systems usability was associated with fewer perceived problems. After controlling for systems usability, a 1 SD increase in food habits score resulted in a 0.274 increase in perceived problems, *B 0*.274 [0.014, 0.521], *p* < 0.05. The systems usability score accounted for a significant 11% of variance, *R*^2^ = 0.11, *F* (1, 53) = 6.53, *p* < 0.05, followed by food habits and mindfulness, which accounted for an additional nonsignificant 9% of variance, *R*^2^
_change_ = 0.09, *F*_change_(2, 51) = 3.10, *p* = 0.054. In combination, the three variables explained a significant 20% of variance, *R*^2^ = 0.20, adjusted *R*^2^ = 0.16, *F*(3, 51) = 4.24, *p* < 0.01, with systems usability and food habits being two significant predictors (Table 5) producing a ‘medium to large’ effect (*f ^2^* = 0.26) [41].

## 4. Discussion

Given the myriad influences on self-report error and the limited research on controlling for this within dietary intake, the current study is one of the few to use the think-aloud methodology to understand participants’ experiences better. As such, we found that participants experienced significantly more total perceived problems while completing ASA24^®^ as compared with INTAKE24© across all problem types (i.e., remembering, program, and emotion). We found that systems usability and food habits were predictive of total problems within INTAKE24©, but no similar association within ASA24^®^.

Previous research [9,10,12] has focused on comparing program methods, features, and reliability or else on demographic and psychosocial factors to explain misreporting errors [44]. A novel aspect of the current study was to use the explorative concept of thinking aloud to better understand participants’ perceived problems during their dietary recall experience. For example, a think-aloud method was previously utilised by French et al. [34] to distinguish the types of problems participants encounter while completing questionnaires or programs [34,36]. As such, more descriptive think-aloud results may arise when participants view ambiguous, frustrating, or memory-provoking situations [45] and are an effective way to evaluate higher-level thinking and individual differences in order for researchers to analyse cognitive processes during a task [26]. Given this, participants’ perceived emotion problems might be due to the influence of emotions on reasoning, attention, memory, and decision making [46]. Consistent with previous research [47], increased frustration or stress may have clouded participants’ memory, impairing free recall. Such results are plausible, as participants in a study by Kupis et al. [44] verbalised their experiences of perceived emotion problems whilst using ASA24^®^, explaining they were due to the program’s duration, confusion, and specificity. It is possible that our findings also captured these emotions, and they may be explained by the repetitive prompts employed to reduce self-report error. Such prompts ask the same question multiple times for different items, are specific and challenging to answer confidently, and often discourage participants from adding all items [44]. This program design differs from INTAKE24©. Similarly, multiple pop-up ads or cookies may irritate people while online browsing, and perhaps repetitive prompts elicit this response. This emotional result is consistent with previous research in a similar young adult population that found multiple pop-up prompts throughout nutrition assessment were frustrating [48].

Furthermore, the results indicate that program usability only predicted the frequency of perceived problems for INTAKE24©, accounting for most of the variance. It may be that memory retrieval was easier in INTAKE24© because the system used greater positive and associative priming pictures [49] rather than the repetitive prompts and questions used in ASA24^®^, reaffirming an early study’s findings on the relationship between priming and memory [50]. For example, INTAKE24© provides a picture of what the meal or food item may look like (for participants to click on the image that most closely represents what they consumed) [9]. In contrast, ASA24^®^ asks where the meal or food item was bought and how it was made or cooked (in writing) before providing pictures for portion size [10]. If this is the case, other dietary recall programs may benefit from the use of similar prompts, pictures, and questions to those employed in INTAKE24©. However, further research is required to identify possible priming influences. Such results are important to capture, as participants who found the program frustrating may be less willing to undertake future 24HRs. Possible improvement may involve changes to the pop-up question design (i.e., using a design more like INTAKE24©) so that duplication does not elicit unwarranted emotional responses. This approach may help to increase participant engagement and satisfaction, thereby enhancing recall reliability [44].

In many interventions for dietary recall, participants are asked to recall specific information from their memory whilst learning a new program. As we saw an overall greater number of perceived problems across ASA24^®^, it may be that the program had a heavier cognitive load than INTAKE24©. In line with previous results from Camos and Portrat [51], increasing the cognitive load reduces immediate recall. Garrett [52] discovered that those who learned to use a program via instructional video before completing a task could manage their cognitive load more successfully after accessing the video. Lower cognitive load may have been present when using INTAKE24©, for which participants were provided with a four-minute video tutorial as part of the standardised instructions. In contrast, ASA24^®^ had a self-guided ‘quick tour’. Past research has found that students prefer video instructions over text instructions [53], with video instructions associated with significantly faster progress and fewer errors than no instructions or text instructions [54]. Aligning with the current study’s student sample, this could explain participants’ preference for INTAKE24© over ASA24^®^. Given the study results in combination with previous literature [52], future research could look at implementing access to programs before the commencement of experimental conditions. Such access could have participants practice using the programs before the study to lower cognitive load and thus associated perceived problems. Similarly, it may benefit ASA24^®^ to incorporate a short video tutorial onto their website for participants to view before the recall task.

Mindful eating was not predictive of fewer perceived problems during either recall program. These findings do not align with the results of Higgs and Donohoe [24], who demonstrated that greater mindfulness whilst eating predicted more accurate recall and behaviour change. However, it is likely that Higgs and Donohoe [24] were able to draw these conclusions because of the inclusion of control groups in their experimental design. Another possible explanation may be the experience of stress, inattentive eating, and time constraints in the study’s student sample [55]. Given that mindful eating is heavily reliant on paying attention [56,57] participants who may have eaten whilst studying or in a rush (e.g., if their attention is primarily focused elsewhere) could have incurred more perceived problems in the ‘remembering’ category. Furthermore, we may see this in conjunction with altered stress levels throughout the semester, as participants’ stress levels may have influenced their emotions [58]. Additionally, stress influences attentional processes, contributing to memory distortions [59]. Whilst much of the literature on stress and attention surrounds recollection failures in eyewitness testimonies, it is essential for research to explore how inattentive eating and stress are present in university students [60]. Future studies using a student sample may benefit by requesting further contextual information on how they consumed food (e.g., on the go or sitting down). Additionally, the Depression, Anxiety and Stress Scale (DASS-21) [60] and mindful eating questions could be combined in this context to understand whether stress contributes to inattentive eating and inaccuracies in dietary recalls.

Greater food habits (i.e., more consistent dietary behaviours) accounted for significantly more perceived problems for INTAKE24© only. This result was not the same for ASA24^®^, which is likely due to ASA24^®^ displaying greater variation within perceived problems. As such, food habits are a possible influential factor in assisting dietary recall; however, other factors likely contribute to its influence, and these appeared more salient for ASA24^®^. Although it is unclear why we obtained differing results for food habits between the two programs, a possible reason could be due to the automaticity of habits. To elaborate, this idea argues that repetition begins to produce action without conscious thought (i.e., requiring no attention), thus resulting in either not recalling the intention of the behaviour or no memory of completing the automatic behaviour [61]. In contrast, this result may simply be explained by university students’ general lack of a regular diet, as the standard deviation for the habit score was very large. Obradovic et al. [62] found that students’ dietary habits significantly varied according to their year level, gender, course type, and age. Platania et al. [63] and Tam et al. [64] explain that many students are just beginning to develop independent dietary choices and are influenced by a variety of factors (e.g., the time between work/study lifestyle, costs, living situation, and on-campus food). Although the student population was deemed most appropriate for the current study, its use limits generalisability to the broader population. Previous research using the Creatures of Habit Scale in the general and student population has been performed; however, there have not been any studies looking at the food questions only. Thus, it is difficult to determine whether our sample was atypical or homogenous. Replicating this study in other populations and across a more extended period (e.g., during the semester and outside of semester or exam time) might help highlight whether irregular food habits are an influencing factor. Purposive sampling amongst individuals who actively maintain habitual dietary habits might also provide different results. 

Other studies report conflicting results on the association between repetition and recall [18,19], with inconsistency possibly being due to the amount of repetition. It may be possible that an intermediate level of habit is more easily recalled. As in the Yerkes–Dodson law, an inverted-U repetition model could be a possible explanation [65]. In the future, including groups of different habit levels (e.g., high, medium, and low) could provide more insight into whether a bell-curve distribution can further explain the ambiguity in the literature between habitual learning and memory retrieval. For example, if an individual’s coffee intake is highly habitual, they may not differentiate between the number of cups they have each day. However, if they go to a barbeque at a friend’s house once a week, they may be more likely to recall what was consumed. Although repetitive, the barbeque deviates from a regular daily routine such as coffee consumption. The inverted U has been evidenced in age groups like that in the current study [66], and it was observed that explicit memory retrieval was best when intermediate cortisol levels (i.e., our primary stress hormone) were dosed; however, this has not been explored regarding implicit memory. Incorporating future measures to understand how habitually individuals consume a specific food item or meal might help further understanding of whether, at a particular level of habitual routine, habits are a factor involved in dietary recall.

Besides the novelty of using the think-aloud method, this study demonstrated that this method can be used to provide rich insights into the 24HR process. Such insights are highly valuable to dietary researchers working to detect associations between perceived problems and the hypothesised causes of the problems. Another strength includes the within-person randomised crossover design, making the comparisons between the two programs highly valid. Likewise, a data-driven checklist better defined the studied sample (e.g., our sample involved university students; the general population might have perceived problems differently). Thus, future studies are recommended to incorporate the think-aloud methodology to understand participants’ experiences further and create a checklist derived from their data that best represents their sample.

The study does not come without limitations. Our sample was made up of university students and had significantly more females than males, limiting the generalisability of the findings. Given that our sample consisted of psychology students, the types of measurements used (i.e., Creature of Habit Scale; Mindfulness Eating Questionnaire) might have been influenced by their undergraduate knowledge, thus likely biasing our findings. Similarly, introspection may come easier for an undergraduate psychology student learning these practices as part of their curriculum, likely influencing the impact of thinking aloud on their recall [67]. As such, the study’s methodology would need to be repeated in different age and demographic groups to determine whether the experiences of the two programs are repeated. Another limitation is that instances of misreporting of participants’ intake were not specifically analysed. It could be hypothesized that fewer frustrations experienced when recording dietary intake might reduce the likelihood of dietary misreporting. However, the converse may also be true, in that the food list may not be comprehensive (i.e., limited inclusion of culturally diverse foods or portion sizes). Future analyses could be conducted to explore possible associations between the number of problems and reported energy intake to assess misreporting.

Furthermore, data on participants’ ethnicity were not collected and may have influenced the findings if international or authentic foods (e.g., non-Australian-traditional ingredients) were unavailable in the program. Although we aimed to control the issue via a think-aloud method, self-report error is the most significant factor influencing accuracy in dietary recall [68]. However, there is minimal consideration in the literature regarding individual differences in articulating thought processes during think-aloud tasks (e.g., participants who struggled to think aloud when this was unnatural for them and thus provided inaccurate data). Although pilot testing was beneficial in explaining some differences and the creation of a checklist, future research may benefit from post-program interviews to control reactivity in participants who may have struggled to articulate their thoughts or actively looked for problems.

## 5. Conclusions

The study aimed to compare the perceived problems encountered in two commonly used self-administered 24HR programs (INTAKE24© and ASA24^®^) and explore whether mindful and habitual eating is associated with perceived problems during dietary recall. In a group of predominantly female undergraduate students, INTAKE24© had the least number of perceived problems during 24HR and was the most preferred to use. This may suggest that design of dietary recall programs for ease of use may be associated with less problematic recall experience. However, further questioning to fully understand individuals’ experiences should be considered. Systems usability and food habits were significant predictors of perceived problems when completing INTAKE24©. This reinforces the literature pertaining to habit automaticity [61], cognitive load, and memory [51] and tells us that attempts should be made to account for food habits when using a 24HR design.

## Figures and Tables

**Table 1 nutrients-14-04281-t001:** Examples of Think-Aloud Quotes for each Perceived Problem Category and 24HR Program.

Perceived Problems	ASA24^®^	INTAKE24©
**Remembering** Consumption Portion Item Guessing	“Brand..pfff..don’t know because I don’t know brand, I never remember them… I’ll click any brand…” “Oh… uh, what did I have yesterday for lunch… I didn’t eat food here I…, I had, so I had leftover meals, so it was… [unrelated talking] … I had brown rice” “I forgot there was something that was in there as well… oh is that what it was? Nah that’s probably about it.” “Oh, nah okay, maybe that’s not the left way to… I’ll go cups, I probably had a cup.” “I ate 3 or 4 slices… I’ll just put 3” “I keep forgetting things, what did I have?”	“Jesus, what did I have?” “Urgh… um, ah, let’s say 4” “Quinoa, peas, capsicum… what else did I ate… CMON! Um…” how many of those did you have… oh s##t um…” “I had 2 cups, no what am I talking about I just had 1” “I had butter biscuits, oh, no, there is a specific name umm, butter, puff pastry?”
**Program** What to input Where to input Individual problems	“Mmmhmm, and then do I add my drink that I drunk too? Coz, I drunk a medium Fanta.” “I’ll just write bites, but they were like jalapeno and mac and cheese bites, I can’t find them on the computer…. “ “What should I say it is? Should I just say it’s a pie? Or see if it’s like bites” “It was Cava? Cava? I’ve never had…. Oh, it’s like a potatoey, Indian potato thingy, cava? Is it K? Kava? Can’t find what I’m looking for, maybe type in Indian?” “can’t find what I’m looking for, is that how to spell it?” “What else was in it…what is that cabbage that’s fermented… Kimchi!” “I don’t know if you need that, but I’m going to put it in anyway” “Where can I find chocolate [tuts]……uhhh… can’t find what I’m looking for…” “I wonder if I should add it in… that would be a lot”	“I had an oatmeal biscuit… I wonder if I have to write that specifically?” “I had um, a piece of um like, raw caramel slice… I’ll just say caramel slice because I think argh it’s like a healthy caramel slice [laughs] I want that to be known, its healthy” “Ummm… what time was that? 8? 9? 7:20? 8:30 I had a chicken burger, I guess I probably have to put that in 1 item at a time…um…oh yeah, I’ll do that…” “Bacon…oh…uh, they want them separate, no, go back…”
**Emotion** Frustration Confusion	“Mmmmm. You must add one food before you hit finish your meal. Oh, I thought I did. Why did I not. Okay its. Subway, did I search it or like…” “I don’t know what like, it says what size was it? I don’t know what like a half of, I think, it wouldn’t be a litre, it will be millilitres, oh I don’t know what millilitres are, sorry. But it wasn’t, it was like, [frustrated clicking] why would it only let me pick 10 litres of bourbon and cola like, I’d be comatose.” “Uhhh… breakfast yesterday… I had it quite early, so it was around about 6 am, yeah, uhhhh… oh between 10:30 and, oh, um… I had it quite early though?… oh, whatever yeah, I’ll just do that…” “Oh wow, this is verrry specific” “Chocolate flavoured jellybaby… NO! what the hell is that?” [large sigh when struggling to find an item] “arghhhhhhhh…!”	[big breath in when unable to find the brand] “Chickpeas… ooh um, … lentils?” “Oh s##t ummm…” “Oh, I don’t know how to spell it” “Oh wait, no I saw it before, was it Greek yoghurt or Greek-styled yoghurt. It was a dark blue and white container I got… what’s the difference?

**Table 2 nutrients-14-04281-t002:** Means (*M*) and Standard Deviation (*SD*) Scores for Male (*N* = 14), Female (*N* = 41), and Total (*N* = 55) Participants’ Height, Weight, Age, BMI, and Combined Perceived Problems.

Demographics	*n*	%	*M* (*SD*)		
			Total (*n* = 55)	Male (*n* = 14)	Female (*n* = 41)
Height (cm)			167.7 (8.1)	175.9 (3.9)	164.9 (7.5)
Weight (kg)			69.7 (15.5)	82.2 (11.1)	65.4 (8.1)
Age			25.5 (8.2)	35.5 (11.4)	31 (8)
Body Mass Index (BMI, kg/m^2^)			24.7 (4.9)	26.6 (2.9)	24.1 (1.3)
Underweight	2	3.6	17.5 (.70)	n.a	17.5 (.70)
Healthy Weight	32	58.2	21.8 (1.7)	22.4 (1.6)	21.7 (1.7)
Overweight	10	18.2	26.2 (.70)	26.8 (.8)	26.0 (.6)
Obese	11	20	33.0 (2.4)	34.9 (2.9)	31.9 (1.3)
Combined Perceived Problems	55	100	50.3 (25.4)	46.3 (18.3)	51.6 (27.5)
Remembering	55	100	30 (16.3)	26.3 (12.0)	31.3 (17.5)
Program	55	100	9.1 (7.5)	8.3 (5.6)	9.4 (8.1)
Emotion	55	100	11.1 (9.1)	11.7 (9.9)	10.9 (9.0)
Variable Scores					
Food Habits	55	100	53.2 (11.6)	51.5 (9.2)	53.7 (12.3)
Mindfulness	55	100	55.7 (7.2)	54 (6.8)	56.3 (7.4)

BMI cut-offs: Underweight ≤18.49 kg/m^2^, Healthy Weight 18.5 to 24.9 kg/m^2^, Overweight 25 to 29.9 kg/m^2^, and Obese ≥30 kg/m^2^ [29]. Combined Perceived Problems includes the sum of INTAKE24© and ASA24^®^ perceived problems for each category. N.a refers to Not applicable.

**Table 3 nutrients-14-04281-t003:** Bivariate Correlation Coefficients to Determine Significant Relationships for all Target Variables (*N* = 55).

Variable	1	2	3	4	5	6	7	
**1**. Age	-							
**2**. BMI	**0.35 ****	-						
**3**. Total Mindfulness Score	0.07	−0.23	-					
**4**. Total Habit Score (Food Only)	**−0.33 ***	−0.10	−0.26	-				
**5**. Program Preference	0.14	−0.02	−0.18	−0.23	-			
**6**. Overall System Usability (2)	0.09	0.10	**−0.34 ***	−0.00	**0.45 ****	-		
**7**. Overall System Usability (1)	−0.04	0.23	−0.27	0.08	**−0.35 ****	0.20	-	
**8**. Overall Perceived Problems (2)	0.08	−0.10	**−0.27 ***	0.20	−0.01	0.18	−0.07	
**9**. Remembering Perceived Problems (2)	−0.01	0.07	−0.17	**0.30 ***	−0.01	0.26	−0.01	
**10**. Program Perceived Problems (2)	0.09	−0.08	−0.21	−0.03	0.06	0.09	−0.20	
**11**. Emotion Perceived Problems (2)	0.16	0.20	−0.23	0.02	−0.06	−0.05	−0.02	
**12**. Overall Perceived Problems (1)	0.01	−0.05	−0.07	**0.27 ***	0.22	0.00	**−0.33 ***	
**13**. Remembering Perceived Problems (1)	−0.09	−0.01	−0.05	0.21	0.13	0.01	−0.18	
**14**. Program Perceived Problems (1)	0.04	−0.13	−0.04	0.12	0.22	−0.05	**−0.38 ****	
**15**. Emotion Perceived Problems (1)	0.15	−0.02	−0.07	**0.28 ***	0.18	0.03	**−0.28 ***	
	**8**	**9**	**10**	**11**	**12**	**13**	**14**	**15**
**8**. Overall Perceived Problems (2)	-							
**9**. Remembering Perceived Problems (2)	**0.86 ****	-						
**10**. Program Perceived Problems (2)	**0.69 ****	**0.44 ****	-					
**11**. Emotion Perceived Problems (2)	**0.56 ****	0.16	0.21	-				
**12**. Overall Perceived Problems (1)	**0.56 ****	**0.55 ****	**0.48 ****	0.12	-			
**13**. Remembering Perceived Problems (1)	**0.44 ****	**0.52 ****	**0.34 ***	−0.00	**0.81 ****	-		
**14**. Program Perceived Problems (1)	**0.37 ****	**0.37 ****	**0.41 ****	−0.00	**0.72 ****	**0.28 ***	-	
**15**. Emotion Perceived Problems (1)	**0.44 ****	**0.29 ***	**0.37 ****	**0.33 ***	**0.72 ****	0.24	**0.65 ****	-

* *p* < 0.05 (two-tailed). ** *p* < 0.001 (two-tailed). Program preference is whether participants chose ASA24^®^ (2) or INTAKE24© (1) as their preferred recall program.

**Table 4 nutrients-14-04281-t004:** Program Preference (%) and Results of Paired-Samples t-Test, including Means (*M*), Standard Deviations (*SD*), and Significance Values (*p*) for the Differences Within and Between Problem Categories (Remembering, Program, and Emotion) and Systems Usability for ASA24^®^ and INTAKE24© (*N* = 55).

Variables	ASA24^®^	INTAKE24©
	Total	
Program Preference	32	68
	*M* (*SD*)	
Systems Usability	78.8 (14.6)	82.7 (11.1)
Paired-Samples *t*-Test		
Mean Differences in Perceived Problems Between ASA24^®^ and INTAKE24©		
Overall Perceived Problems	33.1 (17.4) **	17.2 (11.3) **
Remembering	19.4 (11.4) **	10.6 (7.3) **
Program	6.1 (5.5) **	3.1 (3.4) **
Emotions	7.6 (6.8) **	3.6 (4.2) **
Mean Differences in Perceived Problem Categories Within ASA24^®^ and INTAKE24©		
Remembering*Program	13.3 (10.2) **	7.5 (7.1) **
Remembering*Emotion	11.9 (1.7) **	7.0 (7.5) **
Program*Emotion	1.5 (7.8)	-.51 (3.2)

Systems usability and program preference are scored in percentage (%). ** *p* < 0.001 (two-tailed).

**Table 5 nutrients-14-04281-t005:** Standardised (β) and Unstandardised (*B*) Regression Coefficients and Squared Semi-Partial Correlations (*sr^2^*) for Variables Predicting Overall Perceived Problems Associated with ASA24^®^ and INTAKE24© During 24-hour Dietary Recall Using Hierarchical Multiple Regression Analysis (*N* = 55).

Program	*B* [95% CI]	β	*sr* ^2^	*p*
ASA24^®^				
Model 1				
Constant	16.58 [−9.1, 42.5]	-	-	0.200
Systems Usability	0.21 [−0.11, 0.53]	0.18	0.03	0.200
Model 2				
Constant	36.39 [−27.7, 100.5]	-	-	0.260
Systems Usability	0.13 [−0.21, 0.47]	0.11	0.01	0.434
Food Habits	0.22 [−0.20, 0.64]	0.15	0.02	0.295
Mindfulness	−0.46 [−1.2, 0.25]	−0.20	0.03	0.201
INTAKE24©				
Model 1				
Constant	45.10 [23.0, 67.12] *	-	-	0.001
Systems Usability	−0.34 [−0.60, −0.07] *	−0.33	0.11	0.014
Model 2				
Constant	43.50 [3.18, 83.73] *	-	-	0.035
Systems Usability	−0.39 [−0.65, −0.12] *	−0.38	0.13	0.005
Food Habits	0.28 [0.01, 0.52] *	0.27	0.07	0.039
Mindfulness	−0.15 [−0.57, 0.26]	−0.10	0.01	0.465

*B* = unstandardised regression coefficient, CI = confidence interval, β = standardised regression coefficient, *sr*^2^ = % variance uniquely explained by each predictor, * *p* < 0.05 (two tailed).

## Data Availability

The data presented in this study are openly available in OpenScience Framework at https://doi.org/10.17605/OSF.IO/4WMHC.

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
