# Peer review of "Predicting Perceived Problems in Self-Administered 24-Hour Dietary Recalls: A Quantitative Think-Aloud Study Comparing Automated Self-Assisted 24-Hour Dietary Assessment Tool (ASA24®) and INTAKE24© in University Students"

_nutrients, 2022, doi:10.3390/nu14204281_

Round 1

Reviewer 1 Report

Overall the study is well done and the paper is thorough. I have no major concerns. The paper addresses a gap in the literature and sheds light on how factors influence reporting in automated systems.

Line 259, sentence starting with "Past validation from ..." is unclear.

Recommend revising Table 3 to a landscape format for easier interpretation.

Reviewer 2 Report

Overview

The article by Mackenzie et al. is a well-written manuscript that offers an interesting, interdisciplinary, and innovative focus on the limitations of 24-hour dietary recall programs. However, this manuscript requires several improvements. My main comments are about the lack of clarity in the objectives and the methods and errors in the referencing (bibliography). Globally, this manuscript is quite long for a nutrition journal. The Introduction and Discussion could be shortened and messages more directly presented/discussed in my view.

Abstract

Very clear and well-written.

A randomised quantitative cross-over design was used, which is a real plus of this study. This should be stated in the abstract.

Introduction

The first two paragraphs are clear and well-written.

The third paragraph has a few words that are bold for no reason.

The fourth paragraph should be reformulated because the general message is unclear, especially in sentences related to lines 80-82 and 87-90.

In addition, there are several problems with referencing (e.g., reference 22 ≠ Rodero et al., no reference 24, no reference 26). Please check the references for the whole manuscript.

Overall, be more straight to the point.

Study objectives

There is a discrepancy between the objectives stated in the abstract and main text (= to compare the perceived problems encountered in two commonly used self-administered 24HR programs and explore if mindful and habitual eating is associated with perceived problems during dietary recall) and the objectives stated in the conclusion (=to assess whether habits and mindfulness could predict perceived problems raised during completing both 24HR programs and if these perceived problems aligned with program preference). Throughout the manuscript (esp. Data Analyses (Analysis 2), Results for Analysis 2, and Discussion) you navigate between both objectives without having a clear priority, which leaves the reader a bit confused about the main objectives. Please decide the main objectives of the paper and adapt the chapters (Abstract, Objectives, Data Analyses (Analysis 2), Results for Analysis 2, Discussion, and Conclusions) accordingly.

Methods

There are several typos throughout the manuscript (e.g., in line 134, no space between the word “methodology” and the reference). Also check lines 160, 169, 197, 202, 210, 243 (remove comma), 253 (“considered as”), 259, 264 (remove comma), 600 and 602 (no more using the abbreviation 24HR).

You refer to three different references for the think-aloud methodology (ref 29, line 134 and ref 31 and 32, lines 238). Please refer your methodology to one reference methodology or argue why you need several references/methodologies and how methodolgies are different from and/or similar to each other.

Line 178: This is unclear what is meant by “each session”. Is it time to complete the 24HR? Please precise.

Line 179: Please give a money-wise equivalent of what 5 SONA represent.

Line 215-217: A reference to these previous results is needed.

Lines 217-218: Do you refer to internal consistency for routine or automaticity? Or provide 2 alphas if you refer to both. Idem for line 234, it is unclear which aspect(s) has(have) good reliability (awareness or recognition?).

Line 259: Past validation from what? (a word is missing!)

Line 288: The chapter Preliminary Analysis should be named Sample size calculation and should be placed before Assumptions in my view.

Line 292: Power calculation provides one number, not an interval. Please state the exact number (55, 56, 57, 58, 59, or 60?).

Lines 300-301: This sentence is unclear (what do you mean by “cases”?). Please rephrase.

Please define more clearly the intercorrelation analyses (for Table 3) in Data analyses.

Lines 308-309: what do you mean by “a series of related samples t-test”?

Results

Line 326: Use “main” study instead of “official”?

Lines 350-351: This is unusual to refer to Table 4 before introducing Table 3. I would therefore delete this part of the sentence à full stop after “Table 2.”

Table 2: one digit after the comma for all numbers.

Table 2: the mean (SD) BMI of underweight males is not 0(0). I suppose this should be “Not applicable” because no men were underweight.

Lines 381-385: 1 digit after the comma. This chapter also does not give information about the within differences (end of Table 4). Please do so. Similarly, the chapter Data analyses do not refer to the methods used for this analysis (e.g., what is meant by “x”? an interaction term?”) or it is unclear.

Table 4: Use separate lines for Program Preference, which is a true percentage of a categorical variable and for Systems Usability, which is a mean (SD) of a “continuous” variable ranging from 0 to 100.

Table 4: “Mean Differences in Perceived Problem Categories Within ASA24® and INTAKE24©” is not aligned to the left whereas “Mean Differences in Perceived Problems Between ASA24® and INTAKE24©” is.  

Line 406: This method by Cohen should be explained in the Methods. This is uncommon to include a method and a reference to this method in the Results.

Lines 438-439: Differences in the font sizes of the writing.

Discussion

Overall, the discussion should be more concise. Please be more straight to the point, especially in paragraphs 3 and 4. Having paragraphs 3 and 4 shorter would allow for discussing other possible explanations for preference for INTAKE24.

First paragraph: Well written. Be sure this is in line with the new objectives.

Lines 452-455: I am not fully sure we can compare the percentage of problems raised during the completion of a 30-min 24HR vs. a short? general questionnaire. Please discuss this.

Lines 523-524: Out of scope. Please delete this sentence.  

Lines 530-531: Clarify if this is a positive (or negative) association. As stated here (and without reading results), this is unclear if having more consistent dietary habits is associated with more or less perceived problems.  

Lines 534-538: Unclear. Please rephrase this sentence.

Lines 553-569: interesting.

Line 585: Define or rephrase “authentic”?

You had also information about food group intake, total energy intake, and other nutrition intake information. Did you analyse them to estimate under-reporting using both 24hR programs? If yes, please provide some results. If not, argue and write this in the limitation sections.

Supplement

Figure S2. “Recognition subscale” instead of “Routine subscale”?

Figure S2. Mind_R_6 is not aligned to the left and with larger font size.

References

Please check and amend references 30 and 31.

Reviewer 3 Report

I thought this was a really interesting paper - and the use of 'think aloud' is a novel and interesting idea (in the context of its potential to help improve dietary survey methodologies).

I only really have a couple of (minor) comments:

1. The authors do comment about the limitations of the type of participants (students) in this type of study (used for convenience.  I think this could be taken further, since the some of the types of methodologies used (such as Creature of Habit Scale, Mindfulness Eating Questionnaire etc.) might come across as more intuitive or familiar to psychology students than perhaps students studying different types of academic disciplines).  It might be that there's something about 'Think Aloud' itself - which is easier for psychology students to work with (than it would be for other students, or other types of participants).  It might be worth some additional comments about limitations of the participants in this study.

2. I'm not familiar with 'think aloud' - but I'm not sure if the paper needs a little more about the practical challenges of using this in practice?  I'm not sure if researchers already using this have different approaches to using this methodology, or if there's any literature about levels of training needed (or not) - or if there's any thing about consistency/reliability/validation etc.  I wasn't sure if either section 2.4.4 or other sections of the paper (introduction or Discussion) still need a little more about the challenges of using 'think aloud' in practice.  

Some very novel and interesting work though.

Round 2

Reviewer 2 Report

The article by Mackenzie et al. has largely improved.  However, this manuscript requires a few minor changes.

·       Lines 55-56: problem of formatting.

·       The fourth paragraph has a formatting problem; some sentences appear twice.

·       There are still several problems with referencing (e.g., reference 23 ≠ Higgs, no reference 22, 24, and 26 found in the text). Please check the references for the whole manuscript.

·       Formatting issue with Table 3 and chapter 3.4 (should be not be in landscape format)

·       Lines 503-504: not possible to read the text
